# Legal Regulations and the Anticipation of Moral Distress of Prospective Nurses: A Comparison of Selected Undergraduate Nursing Education Programmes

**DOI:** 10.3390/healthcare10102074

**Published:** 2022-10-19

**Authors:** Karen Klotz, Annette Riedel, Sonja Lehmeyer, Magdalene Goldbach

**Affiliations:** Faculty of Social Work, Education and Nursing Sciences, University of Applied Sciences, Flandernstr. 101, 73732 Esslingen, Germany

**Keywords:** ethics, nursing ethics, moral distress, nursing education, ethics training, higher education, ethical competence, nursing law

## Abstract

Moral distress is commonly experienced by nurses in all settings. This bears the risk of a reduced quality of care, burnout and withdrawal from the profession. One approach to the prevention and management of moral distress is ethical competence development in undergraduate nursing education. Profession-specific legal regulations function as a foundation for the decision on the educational content within these programmes. This theoretical article presents the extent to which legal regulations may open framework conditions that allow for the comprehensive preparation of prospective nurses to manage moral distress. The legal frameworks and the immediate responsibilities regarding their realisation in the context of undergraduate nursing education vary slightly for the three chosen examples of Switzerland, Austria and Germany. While an increased awareness of ethics’ education is represented within the nursing laws, no definite presumption can be made regarding whether undergraduate nursing students will be taught the ethical competencies required to manage moral distress. It remains up to the curriculum design, the schools of nursing and instructors to create an environment that allows for the realisation of corresponding learning content. For the future, the establishment of professional nursing associations may help to emphasise acutely relevant topics, including moral distress, in undergraduate nursing education.

## 1. Introduction

It is widely accepted that moral decision-making is a fundamental component of professional nursing care which cannot be avoided but needs to be dealt with by healthcare professionals [1,2,3,4,5,6,7,8,9,10,11]. The professional responsibility of nurses to act in a way that is morally appropriate arises from the vulnerability of the individuals to whom they deliver nursing care, and who should be the primary focus of their nursing practice [7]. In order to facilitate moral decision-making and support nurses in their everyday moral practice, professional Codes of Ethics provide a practice direction for nurses [7,9,12,13,14]. Since there has been an increase in ethical quandaries [15,16,17,18] and ethically complex situations in the contemporary context of nursing [2,19,20,21,22] it is crucial that prospective nurses develop the appropriate ethical competencies that enable them to uphold the moral virtues, duties and principles which are set in such professional standards. For this reason, there should be an emphasis on ethics education and ethical competency training for nurses early on in their careers, starting in the undergraduate nursing education programmes [15,23,24,25,26]. This may guide nurses through the process of developing comprehensive ethical competencies including ethical knowledge [6,24], mindfulness [24,27], moral sensitivity [28,29,30], moral resilience [24,27,31], moral courage [24,26], professional moral values [24], communication skills and guidance in the process of breaking bad news [32,33], as well as ethical reflection [6,34]. These abilities may support prospective nurses in maintaining their moral integrity, understood as the “moral wholeness” (p. 78) [35] of a person or as their ability to live up “to one’s personal moral code, so that one can sleep at night, or live with oneself, having demonstrated courage, patience, and perseverance in the face of conflict” (p. 431) [36]. If a person’s moral integrity has been violated, this infringement may result in moral distress, a phenomenon of growing interest within the recent nursing literature [2,19,37,38,39,40,41,42,43,44,45,46,47,48,49,50,51,52,53,54].

While the exact number of nurses experiencing moral distress are not known, some authors describe it as a widespread phenomenon [4,19] which has been reported in a variety of care settings internationally [17,55,56,57,58,59,60,61,62]. It seems important to point out that any everyday care situation that gives rise to a moral question may result in the individual’s experience of moral distress [15,19,40]. However, some more extreme care settings or particularly challenging nursing activities, such as the autonomous participation in the process of breaking bad news (further examples will be provided in Section 3.3. ‘Contributing factors’), may carry an increased risk of (moral) stress, compassion fatigue and burnout [32,33]. One recent challenge affecting nurses in all care settings is the COVID-19 pandemic which has been linked to a particularly high prevalence of moral distress among healthcare workers [42,43,44,45,46,47,63]. In view of the multiple (ethical) quandaries that the pandemic brought into healthcare, and the profound negative consequences that may arise from moral distress, including the experience of negative emotions, the violation to one’s moral integrity and a reduced quality of nursing care (further details will be provided in Section 3.2. ‘Consequences of moral distress’), it seems to be of utmost urgency to raise awareness for this phenomenon even if the precise prevalence remains unknown. 

Therefore, moral distress as a relevant phenomenon for nurses’ future professional lives and their nursing practice in the context of undergraduate nursing education will be the subject matter of this article. 

## 2. Aims and Objectives

In this article, we aim to:

(1) Provide an overview of the phenomenon of moral distress, its definitions, dimensions, consequences and the contributing factors to its development in the context of nursing. 

Furthermore, a focus will be put on undergraduate nursing education programmes in regard to this phenomenon. The key question guiding this exploration is the following: to what extent do legal regulations with respect to undergraduate nursing education programmes open up framework conditions to prepare prospective nurses to professionally manage moral distress? In this context, (legal) framework conditions refer to any mandatory laws and orders that prescribe specific ethical content for the nursing curriculum.

The three neighbouring German-speaking countries of Switzerland, Austria and Germany were chosen to be included in the comparison as all three countries are closely linked geographically and have revised their nursing Acts within recent years (the German and Swiss Acts came into force in 2020, the Austrian amendment to the Nursing Act came into force in 2016). Moreover, these three countries have all undergone similar developments and changes in terms of nursing education, with an increasing shift towards academia within the last decade. The legal regulations regarding ethics education, as well as the immediate responsibilities for its realisation will be presented for each of the three nations. This appears to be of interest in order to analyse how ethical education for the prevention and professional management of moral distress may be best included within a new nursing curriculum and how new strategies may be best integrated into the academic nursing education pathway. The academic pathway for general nursing education is relatively new to Germany and with this comparison, the authors expect to derive implications for the ethical competency development of undergraduate nursing students in Germany.

Thus, the further aims of this article include:

(2) To describe, compare and analyse the legal regulations framing ethical education in the selected undergraduate nursing education programmes;

(3) To discuss the implications of the analysis in regard to ethical education, focusing on the prevention and professional management of moral distress of prospective nurses.

## 3. An Approach to the Phenomenon of Moral Distress

### 3.1. Definitions 

If nurses are prevented from doing what they feel is morally right in a situation, and their moral integrity is at stake because they cannot fulfil their professional moral responsibility, they may experience moral distress [64,65], which is used to describe an experience of moral suffering [27]. Various definitions of the phenomenon of moral distress have been discussed within the literature [40,48,49,51,53,66,67,68]. According to Jameton (p. 6) [67], who first described it, moral distress may occur when “one knows the right thing to do, but institutional constraints make it nearly impossible to pursue the right course of action”. Morley et al. [53] defined moral distress more specifically as a direct correlation between the occurrence of a moral event (e.g., moral tension, moral conflict, moral dilemma, moral uncertainty, or moral constraint) and a person’s experience of psychological stress. They point out that the symptoms of such psychological stress may vary among nurses but include negative emotions such as anger, frustration, guilt, regret, sadness, upset, powerlessness, symptoms associated with stress and feeling torn. This description is in line with those of others who point out that moral distress is to be understood as the subjective experience of each individual nurse [37,38,40,41]. Yet another definition of moral distress is that provided by a systematic review by Sanderson et al. [51] (p. 207) who describe moral distress as follows: the “Ethical unease or disquiet resulting from a situation where a clinician believes they are contributing to avoidable patient or community harm through their involvement in an action, inaction or decision that conflicts with their own values”. This definition seems to be supplementing effectively the understanding of moral distress in the educational context of this article, as the ethical unease in this definition results out of the violation of the individuals “own values”. This seems important as there is an explicit distinction to be made between personal and professional values and gaining the appropriate competence to understand and reflect this distinction; additionally, to develop and internalise professional moral values ought to be the subject matter of training in ethical competence in undergraduate nursing education programmes [24,43,69,70,71]. 

Other dimensions of moral suffering such as moral injury [38,72,73,74] and moral residue [38,75,76] are being discussed in the literature with respect to the continuum of moral suffering. This continuum is understood as a person’s oscillation between the feelings of moral comfort (the individual’s moral integrity remains intact) and moral distress (the individual experiences a violation to their moral integrity) [39,40]. Moral residue is understood as the consequence of a person’s repetitive experience of unresolved moral distress in a way that causes a gradual accumulation of stress levels over time, resulting in long-lasting, residual feelings of a compromised moral integrity [38,75]. In respect to the dimension of moral injury, Rushton et al. (p. 121) [41] define this experience as follows: “When moral distress is unrelieved or becomes chronic, or the intensity of it overwhelms a person’s capacity to remain whole, it can lead to more severe forms of moral suffering, such as moral injury”. 

Nevertheless, the main focus of this article will be placed on the dimension of moral distress as this appears to be the first escalation level of the experience of moral suffering. If moral distress is dealt with early on or prevented altogether, nurses may not come to experience more severe dimensions of moral suffering such as moral injury or moral residue. 

### 3.2. Consequences of Moral Distress

Negative consequences that may arise from moral distress are numerous and include emotional stress, such as feelings of frustration and anger [5,76], emotional fatigue, feelings of diminished moral integrity and moral sensitivity, depersonalisation, disengagement or even burnout [65,77,78]. Some studies have found that moral distress may cause nurses to leave their job positions or the profession completely [41,43,72,79]. Moreover, the quality of patient care may be impacted negatively [43,44,80,81,82] due to the feeling of powerlessness, emotional fatigue, depletion and burnout that may limit nurses in their ability to stand up for patients’ needs [83]. Because of these drastic possible consequences of moral distress, it is crucial to foster appropriate competencies to prevent and professionally manage moral distress as early as during the course of undergraduate nursing education programmes [8]. 

### 3.3. Contributing Factors

In their model of the development and effects of moral distress, Goldbach et al. [40] described the development of moral distress as a complex interrelationship between various factors that influence the individual’s unique experience of moral situations. These varying factors have an impact on the way a person acts and then copes with such situations. 

For example, the development of moral distress has been linked to the work environment, and ethical climate of an organisation [40,78,84,85]; organisational structures and frameworks [22,40,72,84,85]; high risk areas such as end-of life care, intensive care, neonatal or paediatric care [27,86]; the ethical complexity of care situations; active or passive demands which are put upon nurses to act in a way that is against their understanding of professional moral values [38]; the professional disposition and moral compass of the individual nurse [40,87]; as well as the ethical competencies and educational background of nurses [87]. 

As these contributing factors to nurses’ experience of moral distress are numerous and complex, a focus will be put upon ethical competency development as a subject matter of undergraduate nursing education programmes. This seems to be important as within the ethically complex contemporary context of healthcare, where nurses see themselves confronted with numerous situations which may violate their moral integrity and result in an experience of moral distress, they need to be well prepared for this work-related burden.

## 4. Description of Legal Regulations

Legal regulations underpinning undergraduate nursing education will be discussed using the examples of Switzerland, Austria and Germany. An overview of the laws and orders in each of the three countries, responsibilities and a brief description regarding their contents with respect to ethical education is provided in Table 1. A brief introduction to European law guiding national legislation will be given. In the discussion, the introduced legal regulations will be compared and analysed focusing on the potential they have in respect of functioning as a foundation for ethical competency development in undergraduate nursing education programmes. 

### 4.1. European Law as a Foundation for National Legislation

A noteworthy element in respect of European law affecting the legal regulations in the three chosen countries is the Directive 2005/36/EC of the European Parliament and the Council of the European Union, which is intended to set up a system for the recognition of professional qualifications (including the recognition of general nursing) in the European Union (EU) and Switzerland [94]. This Directive is one of the consequences of the meetings and agreements of the Bologna process that started in 1999 with the intention of ensuring the comparability of the standards and quality of higher-education qualifications within the European Union [95]. 

Article 31 of the Directive (2005/36/EC) sets the framework conditions and content requirement for general nursing education programmes. According to section 6 (article 31) the “nature and ethics of the profession” are to be addressed in the theoretical instruction of prospective nurses. As this is phrased very broadly, it remains up to the national education frameworks of the member states to determine the specific contents of ethical education for the undergraduate nursing curriculum. What is more, according to Directive 2005/36/EC, there is no regulation regarding the competency level that ought to be achieved by the end of the undergraduate nursing education programme. 

For a more comprehensive comparison of the varying (ethical) competency levels, the European Qualifications Framework (EQF) for lifelong learning (2017/C 189/03) may be used [96]. The focus of this discussion will be the comparison of undergraduate nursing education programmes that will conclude with the award of a bachelor’s degree. The corresponding EQF competency level that is to be reached by the end of each of the three presented undergraduate nursing education programmes would be a “level 6” of the EQF [97]. Specifically, this means that graduates (bachelor’s degree in nursing) need to [96]: -Have “advanced knowledge of a field of work or study, involving a critical understanding of theories and principles”;-Have “advanced skills, demonstrating mastery and innovation, required to solve complex and unpredictable problems in a specialised field of work or study”; -“Manage complex technical or professional activities or projects, taking responsibility for decision-making in unpredictable work or study contexts” and “take responsibility for managing professional development of individuals and groups”.

Achieving this level of competence would be expected of graduates of the three selected undergraduate nursing education programmes in relation to the development of ethical competencies to prevent and professionally manage moral distress. In particular, the third of the above-listed competencies is of relevance to the topic of moral distress. If graduates are expected to manage complex professional projects in order to participate in decision-making on completion of their level 6 education programme, they need to be able to manage morally complex situations that have the potential to develop the individual’s management of moral distress as one component of this competence level. Furthermore, as the EQF serves to support the process of lifelong learning, it may be expected that on completion of a level 6 qualification, nursing students will be provided with the appropriate tools that foster their moral resilience and ability to professionally manage moral distress throughout their careers. 

How far the legal regulations provide a framework to open up the appropriate learning conditions with regards to ethical competencies will be discussed in the following section.

### 4.2. Switzerland 

In Switzerland, there was a new Act on the Regulation of Health Professions (“Gesundheitsberufegesetz”, (GesBG)) decided upon in 2016, which came into force on 1 February 2020 [88]. The Act is applicable to various healthcare professions including nursing, midwifery, physiotherapy, occupational therapy, dietary counselling, optometry, and osteopathy (article 2, section 2 GesBG).

There are three occupational groups of the nursing profession with different educational levels in Switzerland including: (1)Health professionals (”Fachfrau/-mann Gesundheit“)/Care professionals (“Fachfrau/-mann Betreuung“)/Health and Social Assistants (“Assistent/-in in Gesundheit und Soziales“);(2)Nurses (HF, tertiary B) who finish their three-year vocational education programme with a diploma;(3)Nurses (FH, tertiary A) who undergo an academic pathway and at the end of their three-year education programme will obtain the title of Bachelor of Science in Nursing.

While both programmes for nurses (HF and FH) qualify a person to work as a nurse, they differ in their admission requirements and competency profiles at the end of each course. Of interest to this article will be the academic programme (nurses (FH)). 

Numerous competencies that nursing graduates need to achieve by the end of their training are listed within the Act on the regulation of health professions (GesBG). Article 4 summarises social and personal competencies and refers to the ethical competency requirements under section 2. According to this, graduates of all health professions ought to take responsibility for individuals, society and the environment, in line with ethical principles (Article 4, section 2). While this is the only part of the Act (GesBG) that refers to ethical competencies, another order covering the profession-specific competency requirements (“Gesundheitsberufekompetenzverordnung”, (GesBKV)) came into force in conjunction with the Act in 2020 (GesBG) [91]. Article 2 of this order (GesBKV) refers to competencies specific to nursing and, under point (h) (Article 2), states that nurses ought to act in a person-centred, caring manner in accordance with principles of nursing ethics. 

Article 5 of the 2020 main Act on the regulation of health professions (GesBG) determines that the responsibilities regarding profession-specific competencies are to be decided upon by the Swiss Federal Council (Bundesrat) in cooperation with universities, university institutions and the relevant professional unions for each of the health professions. In Switzerland, the professional union for nursing is the Schweizer Berufsverband der Pflegefachfrauen und Pflegefachmänner (SBK-ASI) [98]. Nevertheless, setting and reviewing competency standards remains the responsibility of the Federal Council and the Department of Public Health who are being advised by university institutions and the SBK-ASI. In respect of the academic nursing pathway (nursing FH), the implementation of a curriculum that allows for the achievement of the competencies that are listed in the Act on the Regulation of Health Professions (GesBG) and the Order on the Profession-specific Competence Requirements (GesBKV) remains with the individual university. 

### 4.3. Austria

Recently, there has been a change in the Austrian Nursing Act of 1997 (“Gesundheits-und Krankenpflegegesetz”-(GuKG)) through the 2016 amendment to the GuKG [89]. According to article 1 of this Act, three occupational groups of the nursing profession shall be educated and deployed in Austria. These are:(1)Nursing assistants (“Pflegeassistentin/Pflegeassistent”) who complete a one-year educational programme;(2)Nursing assistants (“Pflegefachassistentin/Pflegefachassistent”) who complete a two-year educational programme;(3)“Nurses of higher service” (“gehobener Dienst für Gesundheits- und Krankenpflegende”) who, according to article 41 sections 1 and 2 of the GuKG, accomplish a three-year academic nursing programme who will be awarded a bachelor’s degree.

The two groups of nursing assistants (“Pflgeassistentin/Pflegeassistent”; “Pflegefachassistentin/Pflegefachassistent”) do not only differ in the length of the education process but also in the level of qualification obtained and the ability (by law) to work autonomously. While the latter are allowed to perform certain tasks independently, nursing assistants undergoing the one-year programme are to be supervised by nurses of higher service. In the past, nursing students were able to achieve that title through the accomplishment of a three-year vocational training programme which was likewise divided up into theoretical and practical hours. However, according to the 2016 amendment of the GuKG, the academic pathway concluding in the award of a bachelor’s degree will be the only possibility of becoming a “nurse of higher service” from 1. January 2024 [99]. 

The legislation of the nursing profession in Austria moves the ethical dimension of nursing in articles 12, 14, 16, 42, 66 and 68 of the GuKG into fore. Article 12 states that ethical perspectives and principles should guide nurses of higher service in their nursing care throughout settings. Article 14, section 2 defines ethical action as one of the core competencies of nursing, while, according to article 16, ethical expertise is ought to be one of the attributes that nurses bring into the multi-professional team in order to participate in ethical decision-making. Articles 42, 66 and 68 each describe educational content for the general nursing education programme (article 42), the paediatric nursing education programme (article 66) and speciality settings (article 68) which all ought to address professional ethics [89]. 

The educational regulations on academic nursing education (“Verordnung der Bundesministerin für Gesundheit, Familie und Jugend über Fachhochschul-Bachelorstudiengänge für die Ausbildung in der allgemeinen Gesundheits- und Krankenpflege- FH-GuK-AV”) serve to complement the Nurses Act (GuKG) and to set the learning outcomes with regards to the core competency development of nursing students during their three-year programme. These regulations were developed and published by the Austrian ministry. There are five annexes to these regulations that list five core competencies and sub-competencies for each of these core learning goals. Ethical competencies appear to play an important role within this document as various ethical competency goals are listed and described under annexes 1, 2 and 4 (FH-GuK-AV) [92]. 

However, while these educational regulations, in conjunction with the GuKG, serve to set the standards and learning outcomes that are expected of graduates on completion of their three-year undergraduate programme, the curriculum is designed by each individual university. An expert committee is then appointed by the Austrian ministry to control whether or not the educational standards as set in the FH-GuK-AV are met by each individual university [100]. 

### 4.4. Germany

In Germany, the latest Act on the Nursing Professions (“Pflegeberufegesetz”—(PflBG)) was decided upon in July 2017 and signed into law on 1 January 2020 [90]. According to the Act, there are two educational pathways available that both qualify a person to work as a general nurse. Prospective nurses may either:(1)Complete a traditional three-year vocational training programme which is comprised of theoretical and practical hours;(2)Choose an academic pathway for a minimum duration of three years, which is comprised of theoretical and practical hours. On achievement, students will be awarded a bachelor’s degree.

Within the German Act on Nursing Professions, ethical education appears to play an important role. Article 5 of the Act (PflBG) summarises competency goals which are expected of students on completion of their nursing education programme: “Throughout their training as a general nurse, trainees develop and strengthen a professional, ethically grounded understanding of the nursing profession and a professional self-image” [90]. This section applies to both the academic and vocational educational pathway. In conjunction with the German Nurses Act, an order on training and examination regulations (“Pflegeberufe-Ausbildungs- und -Prüfungsverordnung (PflAPrV)) came into force in 2020 [93]. Five core competency goals which are required on graduation from the nursing education programme (academic and vocational) are listed under annex 2 of these regulations (PflAPrV, referring to article 9, section 2, clause 2). To be able to act ethically plays a major role within core competency II (person- and situation-centred professional communication and advice) as ethical subgoals for graduates, including for the graduates to:Stand up for the realisation of human rights, codes of ethics and the promotion of the individual needs and habits of their patients and their reference persons;Support the self-realisation and self-determination of people in their care in consideration of possibly conflicting ethical principles;Contribute to the interprofessional discussion and decision-making in regard to ethically challenging (dilemma) situations.

Furthermore, core competencies IV (to reflect and justify one’s actions on the grounds of legal regulations and ethical guidelines) and V (to reflect and justify one’s own actions on the basis of research and professional ethical principles) likewise put a focus on the ethical competencies which are expected of nursing graduates. 

While competency goals for undergraduate nursing education are set within the PflAPrV, the German Act on Nursing Professions (PflBG, article 53) tasks an expert commission with the generation of the framework curriculum and therefore, the first step in the realisation of ethical competency training goals within the academic nursing education programme. The responsibility for the implementation of the framework curriculum and module design then remains with each individual university. Nevertheless, the main control over setting up framework conditions for ethical competency development in the undergraduate nursing education programme remains in the hands of politicians who have the power to appoint the experts to this commission.

## 5. Discussion

The ethical dimension of nursing is explicitly represented at varying levels within the legislation regarding the nursing profession in Switzerland (GesBG; GesBKV), Austria (GuKG; FH-GuK-AV) and Germany (PflBG; PflAPrV) [88,89,90,91,92,93]. First, in the following discussion, the immediate responsibilities in relation to setting appropriate (ethical) competency goals and ethical standards for the undergraduate nursing education programmes will be compared and analysed. Secondly, a focus will be put upon the specific ethical content and the suggested competency promotions which are phrased within the legal regulations on the profession. An overview of these responsibilities and the ethical learning content are provided within Table 1 which may aid the reader to identify the differences and similarities between the three chosen countries.

### 5.1. Education and Professional Associations of Nursing

#### 5.1.1. Comparison

In Switzerland, the responsibilities for setting appropriate (ethical) competency goals and ethical standards for the undergraduate nursing education programme are shared between the Federal Office of Public Health (“Bundesamt für Gesundheit”), the Swiss State Secretariat for Education, Research, and Innovation (“Staatssekretariat für Bildung, Forschung und Innovation”) and the relevant universities, university institutions and the Swiss professional union for nursing (SBK-ASI). The SBK-ASI is a democratic organisation that is, inter alia, tasked with the representation and establishment of the nursing profession, the professional advancement and quality assurance of nursing care, the promotion of professional teaching and research within the profession as well as the active promotion of nursing education (undergraduate and postgraduate education) [98]. According to article 9, section 3 of the GesBKV, competency goals for the undergraduate nursing education programme are to be reviewed every ten years or at an earlier stage if deemed necessary by either of the listed parties [91]. Nevertheless, a partial share of the responsibility regarding setting and reviewing competency goals and standards for nursing and nursing education remains in the hands of politicians. 

In comparison, in both Germany and Austria, the responsibility of setting competency goals lies solely with politicians. These competency goals are summarised in the German (PflAPrV) and Austrian (FH-GuK-AV) training and examination regulations [92,93]. According to the German Act on Nursing Professions (PflBG), article 53, the Federal Ministry for Family Affairs, Senior Citizens, Women and Youth (“Bundesministerium für Familie, Senioren, Frauen und Jugend”) and the Federal Ministry of Health (“Bundesamt für Gesundheit”) are to appoint an expert commission that is responsible for drawing up a framework plan for the nursing education programme. Members of this commission are to have expertise in nursing, nursing education or nursing research (PflBG, article 53, section 3). However, in this framework plan, the expert commission is tied to adhere to the competency goals that are set by politicians in the PfAPrV. In Austria, it is likewise the politicians who set competency goals within the GuKG and the FH-GuK-AV, though there is no commission appointed by law that is tasked with the generation of a framework curriculum. 

#### 5.1.2. Analysis

Giving the full responsibility of setting (ethical) competency goals and (ethical) standards for nursing to politicians, who do not necessarily have comprehensive expertise in the complex field of nursing, seems rather problematic. Nursing is to be understood as a profession that in its complexity involves a high amount of responsibility towards individuals who require nursing care [101,102]. Due to this highly complex nature of nursing, it seems inappropriate and outdated to transfer the responsibility for setting professional standards, competency goals and learning outcomes for this profession to anyone other than members of the profession themselves. Acting as an international role model in this regard are some of the Anglo-American countries, with established professional associations of nursing, that are provided with the full responsibility by law, to set (ethical) standards and competency goals for nursing, including nursing education. As an example, the Irish Nursing and Midwifery Act (2011) delegates the task of setting the appropriate competency goals and professional standards for nursing to the Irish Nursing Association–the Nursing and Midwifery Board of Ireland (NMBI). According to article 85 of the Act: “The Board shall—(a) set and publish in the prescribed manner the standards of nursing and midwifery education and training for first time registration and post-registration specialist nursing and midwifery qualifications, and (b) monitor adherence to the standards referred to in paragraph (a)” (section 1) and “prepare guidelines on curriculum issues and content to be included in programmes approved under paragraph (a)” (section 2) [103]. Similarly, the nursing associations of the individual states of the U.S.A. are responsible by state law, for setting competency goals and standards for nursing education [104]. The Canadian Nurses Association even included the concept of moral distress within their Code of Ethics, prioritising and raising awareness for this phenomenon in the contemporary context of nursing [13]. It is indeed long overdue within the German-speaking context of nursing, to follow this example, to establish such professional nursing associations and to task them with the responsibility of setting evidence-based and up to date (ethical) competency standards for nursing education as a means for quality assurance in nursing [102]. 

### 5.2. Ethical Competency Goals as Prescribed by Law

#### 5.2.1. Comparison

The Swiss Act on the Regulation of Health Professions (GesBG) and its complementing regulations on competency requirements for nursing (GesBKV) broadly mention that graduates of the healthcare professions need to act in line with ethical principles (GesBG, article 4, section 2) and nurses are to act in a person-centred, caring fashion, in line with the specificities of nursing ethics (GesBKV, article 2) [88,91]. These aspects of ethical competency learning outcomes for undergraduate nursing students are phrased very broadly and their interpretation with regards to curriculum design remain up to the individual university. Hence, while a very basic foundation for ethics education in undergraduate nursing education programmes is provided by law, it remains unclear to what extent the phenomenon and professional management of moral distress will be the subject matter of undergraduate nursing education in Switzerland.

In comparison, in both Germany and Austria it appears that ethics education is being weighted rather high within the nursing law, as ethical competency requirements that are expected of (prospective) nurses are listed in various places throughout the Austrian and German Nursing Acts (GuKG: articles 12, 14, 16, 42, 66, 68; PflBG: article 5) [89,90] and the complementary training and examination regulations (FH-GuK-AV: annexes 1,2,4; PflAPrV annex 2, core competencies II, IV, V) [92,93]. While the phenomenon and professional management of moral distress is not mentioned explicitly within these documents, the Austrian FH-GuK-AV and the German PflAPrV implicitly promote the education in relation to this topic and set learning outcomes accordingly. As an example, graduates of the undergraduate nursing education programme in Austria are expected to be capable of weighing up decisions in respect to quandaries which arise because of conflicting interests related to the incompatibility of professional ethics, economic efficiency and the individuality of nursing care (FH-GuK-AV, annex 1). If graduates recognise ethical conflicts, they are expected to voice, reflect and discuss them within the multi-professional team using all of the multi-professional resources available to them (FH-GuK-AV, annex 1). In Germany, nursing graduates are expected to be able to critically reflect and evaluate their care on the foundation of legal regulations and ethical guidelines (PflAPrV annex 2, core competency IV). This requires graduates to be sensitive to moral situations and to be able to ethically reflect these situations subsequently, which can be understood as “moral sensitivity” [105]. 

Within the German training and examination regulations (PflAPrV, annex 2, core competency IV) this reflective process is intended to act as a means of quality assurance. As moral distress has been associated with a reduced quality of nursing care [43,44,80,81,82], providing ethical competency training with respect to this topic within the undergraduate nursing education programme would not only promote the health and well-being of prospective nurses, but also contribute to meeting high-quality care standards. 

Furthermore, nursing graduates in Austria ought to know their own potential as well as their limitations and are required to notice stressors in their work life and use coping strategies accordingly (FH-GuK-AV, annex 2). Similarly, nursing graduates in Germany are expected to make use of appropriate strategies to deal with work-associated burdens and to avail of and insist on supporting measures related to these burdens (PflAPrV, annex 2, core competency V). In order to be able to recognise and manage moral distress as such a work-associated burden in a timely manner, it becomes obvious that prospective nurses need to develop the appropriate ethical competencies to empower them to prevent and manage morally distressing situations. 

#### 5.2.2. Analysis

Riedel et al. [6] suggest that ethical competency in the context of nursing is to be understood as a multifaceted set of professional skills, knowledge, know-how, actions, experience, motivation, attitudes and values. Rabe [106] further identified that ethical reflection was a central element of ethical competency in nursing. Moreover, components of ethical competency in nursing included the ability to voice and to justify one’s own moral values and attitudes; to recognise moral problems; to allow for a change in perspective, problem-solving and the readiness for conflict management; and to take responsibility and show courage in the face of morally challenging situations (p. 207) [106]. It follows that the implementation of educational contents and strategies that allow for the development of the ethical competencies which are set out in the nursing law is a rather complex process, as there is more to ethical competency training than solely transferring knowledge regarding ethical theories. The ICN code of ethics may be viewed as a point of reference or a framework in regard to educating the nursing profession in ethical and moral standards in all of the three presented countries [7]. In Switzerland, guidelines by the Swiss Academy for Medical Sciences serve as a point of reference in regard to ethical training content for all health professions [9]. Accordingly, the goals of ethics education should not be limited to the nursing student’s ability to participate in ethical decision-making but also include the ability to reflect on moral situations and the promotion and protection of their moral integrity in order to foster the health and well-being of prospective nurses. What is more, due to its fundamental nature underpinning all aspects and actions of the nursing profession, ethical competency development should be an area of utmost importance within the schools of nursing and it should be supported, addressed and reflected upon cross-sectionally throughout nursing education [6,24,107]. While legal regulations solely operate as a foundation for ethics education, Riedel et al. [8] strongly suggested that the comprehensive development of ethical competence was a crucial step in the prevention and professional management of moral distress. Nevertheless, while building ethical competence is an important contribution to preventing and managing moral distress, a higher sensitivity for morally challenging situations simultaneously bears an increased risk for the experience of moral distress if it is impossible to relieve the experienced moral unease. Starting to foster moral resilience, understood as “the ability to recover from or healthfully adapt to [moral] challenges, stress, adversity, or trauma” [27] (p. 13) in undergraduate nursing education programmes may counteract this risk in giving prospective nurses the right toolset to professionally manage the experience of moral suffering and to protect their moral integrity. Mandatory group psychotherapy sessions for undergraduate nursing students have been discussed as one innovative curricular activity that fosters both the emotional intelligence needed to develop ethical sensitivity, and resilience as a way of coping with any moral uncertainties that arise in the complex process of nursing care [108]. It may be expected that innovative methods and ideas to strengthen nursing students’ ethical, emotional and coping skills will have to be further integrated into undergraduate nursing curricula.

## 6. Limitations and Future Uses

There are some limitations to this perspective that need further consideration. 

Firstly, only a limited number of undergraduate nursing education programmes were chosen for this contribution. A broader comparison may have provided a deeper understanding of how legal regulations underpinning undergraduate nursing education programmes may open up framework conditions to prepare prospective nurses to professionally manage moral distress.

Secondly, legal regulations underpinning undergraduate nursing programmes simply provide a basic foundation for ethics education while the whole educational system is rather complex. Therefore, it is not possible to draw precise conclusions regarding whether the prevention and professional management of moral distress will be addressed sufficiently in the undergraduate nursing curriculums.

Thirdly, any differences that may arise from a gender point of view with regards to the approach to (moral) distress and conflict situations have not been considered in this work. This may be an interesting perspective which needs further attention in future works in order to discuss whether gender considerations need to play a greater role in the design of nursing curricula. 

Nevertheless, this perspective may help to identify moral distress as a major work-related burden for nurses. The findings can be used to emphasise the importance of the emotional and moral health and well-being of (prospective) nurses as a matter of utmost urgency to prevent nurse burnout and job withdrawals as well as a reduced quality of care.

In an educational context in particular, it is crucial to sensitise the nursing (and healthcare) community to the phenomenon of moral distress due to the increasing ethical demands which are put upon (prospective) nurses and all the morally distressing situations that students may already be confronted with throughout their undergraduate education programme. 

Therefore, the findings of this perspective should be considered by the authorities responsible for the nursing curriculum design and nursing instructors who need to address moral distress and foster appropriate coping strategies in undergraduate nursing education.

## 7. Conclusions

In summary, moral decision-making is fundamental to the nursing profession as patient cases are becoming increasingly complex in the contemporary context of nursing [1,2,3,4,5,6,7,8,9,10,11]. Simultaneously, in all of the three presented countries, the professional requirements and responsibilities of nurses are increasing. The ethical complexity of nursing care bears the risk for the individual nurses’ experience of moral distress which in turn may have negative consequences such as nurse burnout [65,77] and withdrawal from the nursing profession [43,72,92], as well as a reduced quality of patient care [43,44,80,81,82,83]. Hence, it is crucial that nurses are provided with the appropriate ethical toolset to professionally prevent and manage moral distress early on in their careers [6,8,24,25,26]. While ethical competencies appear to play an important role in the introduced undergraduate nursing education programmes, the main focus is being put upon the ability of prospective nurses to provide high-quality patient care which is based on ethically justified decisions and to participate in interdisciplinary ethical decision-making. While these competencies are important to nursing, it seems advisable that a higher focus should be put upon strategies that protect the moral integrity of prospective nurses. This may help to counteract nurses’ development of moral distress by strengthening their moral resilience [8,27].

Only broad presumptions regarding the potential for ethical competency development to prevent and professionally manage moral distress may be drawn from the legal regulations framing nursing education. Even though the exemplary comparison and analysis in this article is limited to three German-speaking countries, it has become evident that ethical competency development throughout undergraduate nursing education cannot be taken for granted. Since ethical competency development may be understood as a crucial contribution to the prevention and professional management of moral distress, the protection of the individual’s (professional) moral integrity, nurses’ job retention and an assurance of the high quality of nursing care, it becomes evident that this is an area which requires further attention in research, nursing education and the nursing community in general. While all of the presented legal regulations provide a broad framework for ethical competency development among undergraduate nursing students, the specific realisation of corresponding training contents remains up to the parties who generate the nursing curriculum, schools of nursing and instructors in nursing education. Nonetheless, this article highlights the importance of addressing moral distress at an early stage and the authors strongly advise the responsible parties for the curriculum design in nursing education and all staff within the schools of nursing to actively promote the development of the comprehensive ethical competencies that may aid prospective nurses in the prevention and management of moral distress. Furthermore, the authors postulate that it is long overdue within the German-speaking context of nursing to establish professional associations for the nursing profession. This would carry the potential for nurses to regulate themselves, strengthening the profession as a consequence. A professional association could better represent the current needs of the nursing profession and put the focus of attention in nursing education on acutely relevant topics such as the experience of moral distress. 

## Figures and Tables

**Table 1 healthcare-10-02074-t001:** Overview of national laws and orders on the regulation of nursing education and references to ethical education contents. (Translated by Karen Klotz).

	Switzerland	Austria	Germany
National Nursing Act	2020 Act on the Regulation of Health Professions (“Gesundheitsberufegesetz”, (**GesBG**)) [88].	2016 amendment to the Austrian Nursing Act of 1997 (“Gesundheits- und Krankenpflegegesetz”—(**GuKG**)) [89].	2020 Act on the Nursing Professions (“Pflegeberufegesetz”—(**PflBG**)) [90].
Supplementing Legal Order	Order on the profession-specific competency requirements (“Gesundheitsberufe-kompetenzverordnung”, (**GesBKV**)) [91].	Educational regulations on academic nursing education (“Verordnung der Bundesministerin für Gesundheit, Familie und Jugend über Fachhochschul-Bachelorstudiengänge für die Ausbildung in der allgemeinen Gesundheits- und Krankenpflege”—(**FH-GuK-AV**)) [92].	Order on training and examination regulations (“Pflegeberufe-Ausbildungs- und -Prüfungsverordnung”—(**PflAPrV**)) [93].
Responsibility for setting competency goals	Partial share of responsibility for setting competency goals remains with politicians.	Full share of responsibility for setting competency goals remains with politicians.	Full share of responsibility for setting competency goals remains with politicians.
References to ethical competence training in undergraduate nurse education programmes	**GesBG [88]:** Article 4, section 2 summarises social and personal competencies and refers to ethical competency requirements. **GesBKV [91]:** Article 2, point (h) refers to competencies specific to nursing.	**GuKG [89]:** Article 12 states that ethical perspectives and principles should guide nurses of higher service in their nursing care throughout settings. Article 14, section 2 defines ethical action as one of the core competencies of nursing. Article 16 states that ethical expertise is ought to be one of the attributes nurses bring into the multi-professional team in order to participate in ethical decision-making. Articles 42 describes educational contents for general nursing education, including nursing ethics. Article 66 describes educational contents for paediatric nursing education, including nursing ethics. Article 68 describes educational contents for specialty settings, including nursing ethics. **FH-GuK-AV [92]:** Annex 1 defines specialist (ethical) competencies for the nursing profession, including individual and organisational competencies. Annex 2 defines social and communicative competencies as well as self-competencies which are based on moral values and ethical reflection. Annex 4 defines minimum requirements, including ethical educational contents for the general nursing training.	**PflBG [90]:** Article 5 summarises competency goals which are expected of students on completion of their nursing education programme, including an ethically grounded understanding of the nursing profession. **PflAPrV [93]:** Annex 2 core competencies II, IV, and V all include aspects of ethics education for undergraduate nursing education programmes.

## Data Availability

Not applicable.

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
