# Peer review of "Legal Regulations and the Anticipation of Moral Distress of Prospective Nurses: A Comparison of Selected Undergraduate Nursing Education Programmes"

_healthcare, 2022, doi:10.3390/healthcare10102074_

Round 1

Reviewer 1 Report

Thank you for this article which explains and compares the legal framework and regulations in nurse undergraduate programs regarding moral distress in tree German speaking countries.

Two of the tree countries have revised their nurse act in the last two years. Furthermore, the Covid-19 pandemic brought rose the awareness of moral distress, which makes this article a relevant contribution.

The choice of countries is pertinent, but needs justification, beyond "It would be beyond the scope of this article..." line 150. This would make sense also regarding the later reference to the Anglo-American countries on page 10, where nurses have a long history of more responsibility and independence from the MD authority than in the studied countries (among other differences).

The structure stated in point 1. Aims and Objectives (describe, compare, analyse, discuss) is clear. According to this, the first part of point 5. Discussion, would fit under "compare", and some later under analyse. 

Beginning with the overarching EU regulation makes sense.

The following description of the three countries is clear. 

Adding a table with the differences and similarities could help to have an overview before the discussion.

A general remark concerning the order of references in [  ]: is it on purpose that they are not in numerical order, nor in chronological order of publication?

Examples in line 36 and 42, 44, 45, 51, 76, 86. Line 51: why 26-28, 29-41 and not 26-41?

line 101: please check: individuals vs individual's?

Author Response

Dear Reviewer,

All authors would like to thank you for taking the time to review our paper and the helpful feedback you provided. We are sure, that your valued advice will help to improve the quality of our paper. We have made changes to the perspective accordingly. Please find our response to the points you made below.

Point 1: The choice of countries is pertinent, but needs justification, beyond "It would be beyond the scope of this article..." line 150. This would make sense also regarding the later reference to the Anglo-American countries on page 10, where nurses have a long history of more responsibility and independence from the MD authority than in the studied countries (among other differences).

Response 1: Thank you for making us aware that our choice of the three selected countries needs further clarification. As you have stated correctly, all three countries have revised their Nursing Acts within recent years. This choice of countries is also of interest to the authors as the academic pathway for general nurse education is relatively new to Germany and with this comparison, the authors expect to derive implications for the ethical competence development of undergraduate nursing students in Germany. You may find our updated explanation of said choice in ll. 88-102.

Point 2: The structure stated in point 1. Aims and Objectives (describe, compare, analyse, discuss) is clear. According to this, the first part of point 5. Discussion, would fit under "compare", and some later under analyse.

Response 2: Thank you for your feedback. We have divided the two main points of the discussion into a section for comparison and analysis in order to make the structure clearer. You may find these changes in ll. 385, 415, 434, 481.

Point 3: Beginning with the overarching EU regulation makes sense.

The following description of the three countries is clear.

Adding a table with the differences and similarities could help to have an overview before the discussion.

Response 3: Thank you for this valuable point. We are certain such an addition would help readers to better comprehend the discussion. We added in an extra row within our table (Table 1) to include a section concerning the responsibility for setting competency goals (point 1 of discussion). We believe that for point 2 of the discussion (‘ethical competency goals as prescribed by law’), the details about differences and similarities are provided within table 1 already (‘references to ethical competence training in undergraduate nurse education programmes’). However, we have added intext references (ll. 186, 379) in order to direct the reader to table 1 to allow for an easier comprehension of the comparisons made. We hope that this will help to better comprehend our discussion.

Point 4: general remark concerning the order of references in []: is it on purpose that they are not in numerical order, nor in chronological order of publication?

Examples in line 36 and 42, 44, 45, 51, 76, 86. Line 51: why 26-28, 29-41 and not 26-41?

Response 4: Thank you for making us aware of this important issue. We have made the appropriate changes to bring our references into numerical order.

Point 5: line 101: please check: individuals vs individual's?

Response 5: Thank you for making us aware of this grammar mistake. We changed the wording into ‘individual’s’.

Again, we would like to thank you for giving your valued opinion on our work.

Best Regards,

The Authors

Reviewer 2 Report

Congratulations to the authors for focussing on one of the most neglected (though important) aspects around moral distress: specific educational policies to prevent its occurrence (and/or acutization).

Substantial improvements should be achieved in terms of quality of the presentation, in order to ensure the impact it deservers.

In general, extensive English revision should be performed to improve fluency, and extensive presentation revision should be performed to lighten the reading (i.e., avoiding repetitions )

Find some suggestions in particular, along with a couple of references that would be important in the background.

ll. 36-37 A couple of the following recent papers (two of which published on this journal) should be cited as they precisely deal with the topic:

10.3390/healthcare9101307

l. 56 remove the point after ) or uniform.

ll. 60, 406 to what extenT

ll. 64, 150-151 why these examples, are they relevant? I think you should specify why these examples are relevant, if they are, or take for granted that a systematic review on international policies not the goal of the present paper, and directly introduce the selected countries

l. 61 it should be clarified what is meant by "opening up framework conditions"

l. 79 defineS

l. 87 describeS

l. 88 ... as THE "ethical unease ..."

ll. 134-136 I recommend to mention this paper

10.7429/pi.2022.75159

l. 153 an overview OF the laws

ll. 256-58 is there a difference beside the additional year of study? It would be relevant to understand (i.e., skills or qualification)

ll. 346-348, 395-397 remove the introducing preposition, it was already specified at the end of the previous paragraph

Author Response

Dear Reviewer,

All authors would like to thank you for taking the time to review our paper and the helpful feedback you provided. We are sure, that your valued advice will help to improve the quality of our paper. We have made changes to the perspective accordingly. Please find our response to the points you made below.

Point 1: In general, extensive English revision should be performed to improve fluency, and extensive presentation revision should be performed to lighten the reading (i.e., avoiding repetitions)

Response 1: Thanks for your valuable feedback. We revised our paper accordingly, paying special attention to the specific points you made (see below). We have also tried to reduce any repetetive components within our work to allow for easier reading of the paper.

Point 2: ll. 36-37 A couple of the following recent papers (two of which published on this journal) should be cited as they precisely deal with the topic (see citavi list provided).

Response 2: Thank you for providing us with some additional literature which we have studied with interest. As you have mentioned, some of the papers precisely deal with the phenomenon of moral distress and provide a comprehensive insight into this important topic. To further enhance the theoretical foundation of our paper, we have added some citations which you may now find in our list of references.

Point 3: l. 56 remove the point after ) or uniform; ll. 60, 406 to what extenT; l. 79 defineS; l. 87 describeS; l. 88 ... as THE "ethical unease ..."; l. 153 an overview OF the laws

Response 3: Thank you for these important corrections. We have revised the grammar and spelling accordingly.

Point 4: ll. 64, 150-151 why these examples, are they relevant? I think you should specify why these examples are relevant, if they are, or take for granted that a systematic review on international policies not the goal of the present paper, and directly introduce the selected countries

Response 4:  Thank you for making us aware that our choice of the three selected countries needs further clarification. As you have stated correctly, all three countries have revised their Nursing Acts within recent years. This choice of countries is also of interest to the authors as the academic pathway for general nurse education is relatively new to Germany and with this comparison, the authors expect to derive implications for the ethical competence development of undergraduate nursing students in Germany. You may find our updated explanation of said choice in ll. 88-102. Furthermore, we have listed in our limitations, that only a limited number of undergraduate nurse education programmes was chosen for this contribution (ll. 521-525).

Point 5: l. 61 it should be clarified what is meant by "opening up framework conditions"

Response 5: Thank you for raising the issue, that this wording needs further clarification in the context of this perspective. We have added in an explanation which you may find in ll. 82, 83.

Point 6: ll. 134-136 I recommend to mention this paper: doi: 10.7429/pi.2022.75159

Response 6: Thank you for making us aware of this rather interesting paper that fits in well within the context of our perspective. We have added in this citation according to your recommendations.

Point 7: ll. 256-58 is there a difference beside the additional year of study? It would be relevant to understand (i.e., skills or qualification)

Response 7: Thank you for drawing our attention to this issue which requires further clarification. To answer this question we have added in some additional information which you may now find in ll. 292-297.

Point 8: ll. 346-348, 395-397 remove the introducing preposition, it was already specified at the end of the previous paragraph

Response 8: Thank you for your feedback. To avoid repetitions, we have limited the outline of the discussion section to the start (ll. 371-380).

Again, we would like to thank you for giving your valued opinion on our work.

Best Regards,

The Authors

Reviewer 3 Report

I would like to thank the editors of this journal for the opportunity to give my point of view on this interesting research project. On the other hand, I would like to congratulate the authors of this research, I believe that it is very necessary to investigate the state of well-being and mental health of nursing professionals and to address this issue from undergraduate studies.

I consider that the study is well thought out and the methodology is appropriate, I would like to make some comments that I believe could be addressed by the authors and improve some aspects of the study.

The introduction would benefit from addressing the most recurrent themes that affect nursing staff stress, such as the communication of bad news. For this I recommend the following bibliography:

https://pubmed.ncbi.nlm.nih.gov/28977239/
https://pubmed.ncbi.nlm.nih.gov/28945712/
https://pubmed.ncbi.nlm.nih.gov/27349844/

Working groups with patients may also be of interest:

https://pubmed.ncbi.nlm.nih.gov/30685106/
https://pubmed.ncbi.nlm.nih.gov/29153455/

On the other hand, I believe that the differences that may exist from a gender point of view should be considered, with a view to establishing differences in the approach to distress and conflict situations in the workplace and, by extension, in education and the design of educational proposals.

https://pubmed.ncbi.nlm.nih.gov/33283488/
https://pubmed.ncbi.nlm.nih.gov/31155275/
https://pubmed.ncbi.nlm.nih.gov/34831556/

I believe that the limitations section is brief and could be expanded, and would also benefit from a section where future uses of the study's conclusions could be proposed.

Once again, I congratulate the researchers for the effort and dedication they have put into this great study and I thank those responsible for the journal for the opportunity to have contributed my opinion on the research.

Author Response

Dear Reviewer,

All authors would like to thank you for taking the time to review our paper and the helpful feedback you provided. We are sure, that your valued advice will help to improve the quality of our paper. We have made changes to the perspective accordingly. Please find our response to the points you made below.

Point 1: The introduction would benefit from addressing the most recurrent themes that affect nursing staff stress, such as the communication of bad news. For this I recommend the following bibliography:

https://pubmed.ncbi.nlm.nih.gov/28977239/

https://pubmed.ncbi.nlm.nih.gov/28945712/

https://pubmed.ncbi.nlm.nih.gov/27349844/  

Response 1: Thank you for providing us with some additional literature which we have studied with interest. We added a paragraph in introduction in order to give a brief overview of the prevalence of moral distress and factors affecting staff stress (such as extreme care situations, including the breaking of bad news, or the impacts of the Covid- 19 pandemic on healthcare (ll.53-68). We have added in some of the citations you recommended which we believe enhanced the quality of our introduction. Furthermore, we added in a line to refer to section 3.3 (Contributing factors), where we briefly describe various factors that influence the individuals unique experience of moral situations.

Point 2: Working groups with patients may also be of interest:

https://pubmed.ncbi.nlm.nih.gov/30685106/  

https://pubmed.ncbi.nlm.nih.gov/29153455/

Response 2: These papers add an interesting perspective into a method that may be used throughout undergraduate nurse education to simultaneously foster emotional skills, moral sensitivity and resilience which are all important competencies for the prevention and professional management of moral distress. We have integrated this idea within our discussion under point 5.2 ‘Ethical competency goals as prescribed by law’ (ll. 513-519).

Point 3: On the other hand, I believe that the differences that may exist from a gender point of view should be considered, with a view to establishing differences in the approach to distress and conflict situations in the workplace and, by extension, in education and the design of educational proposals.

https://pubmed.ncbi.nlm.nih.gov/33283488/ Title: ‘Understanding equality and diversity in nursing practice’

https://pubmed.ncbi.nlm.nih.gov/31155275/  Title: ‘Gender inequality and restrictive gender norms: framing the challenges to health’

https://pubmed.ncbi.nlm.nih.gov/34831556/ Title: ‘Health Inequities in LGBT People and Nursing Interventions to Reduce Them: A Systematic Review’

Response 3: Thank you for this feedback and the recommended literature you provided. We do agree that a gender point of view in relation to this (educational) topic is rather interesting and should be looked at in more detail. As we believe adding this perspective into our proposed paper might extend the scope of this work, we have added this lack of consideration of the gender persective in our list of limitations. We think future works could look at the topic from a specific gender point of view.

Point 4: I believe that the limitations section is brief and could be expanded, and would also benefit from a section where future uses of the study's conclusions could be proposed.

Response 4:  Thank you for this important point. We have expanded our list of limitations and included a section to outline future uses of our paper. You may find these changes in ll. 520- 548.

Again, we would like to thank you for giving your valued opinion on our work.

Best Regards,

The Authors

Reviewer 4 Report

Dear authors,

The above perspective provides an depth analysis of moral distress in nurses. The objectives are clear, concise and prepare the readers for the content. The flow is nice and relevant articles are cited. However, my major concern is in introduction. Relevant statistics from organisations, or  published literature  are missing . A table or a short paragraph should be included in the introduction  to explain with numbers the situation of moral distress in nurses. 

Author Response

Dear Reviewer,

All authors would like to thank you for taking the time to review our paper and the helpful feedback you provided. We are sure, that your valued advice will help to improve the quality of our paper. We have made changes to the perspective accordingly. Please find our response to the points you made below.

Point 1: However, my major concern is in introduction. Relevant statistics from organisations, or  published literature  are missing . A table or a short paragraph should be included in the introduction  to explain with numbers the situation of moral distress in nurses.

Response 1: Thank you for this important point that shows how the relevance of the topic needs further explanation. We have added in a paragraph to outline the relevance of moral distress to nursing in more detail. You may find these changes in ll. 53-68. Particularily in view of the Covid- 19 pandemic and the multiple ethical challeges in brought upon nurses we believe it is of upmost urgency to sensitise the nursing community for the phneomenon in order to prevent this work- related burden.

Again, we would like to thank you for giving your valued opinion on our work.

Best Regards,

The Authors